# Taxifolin Inhibits the Growth of Non-Small-Cell Lung Cancer via Downregulating Genes Displaying Novel and Robust Associations with Immune Evasion Factors

**DOI:** 10.3390/cancers15194818

**Published:** 2023-09-30

**Authors:** Xiaozeng Lin, Ying Dong, Yan Gu, Fengxiang Wei, Jingyi Peng, Yingying Su, Yanjun Wang, Chengzhi Yang, Sandra Vega Neira, Anil Kapoor, Damu Tang

**Affiliations:** 1Department of Surgery, McMaster University, Hamilton, ON L8S 4K1, Canada; linx3636@gmail.com (X.L.); dongy87@mcmaster.ca (Y.D.); guy3@mcmaster.ca (Y.G.); peng.jingyi96@outlook.com (J.P.); suy36@mcmaster.ca (Y.S.); veganeis@mcmaster.ca (S.V.N.); akapoor@mcmaster.ca (A.K.); 2Urological Cancer Center for Research and Innovation (UCCRI), St Joseph’s Hospital, Hamilton, ON L8N 4A6, Canada; 3The Research Institute of St Joe’s Hamilton, St Joseph’s Hospital, Hamilton, ON L8N 4A6, Canada; 4The Genetics Laboratory, Longgang District Maternity and Child Healthcare Hospital of Shenzhen City, Longgang District, Shenzhen 518174, China; haowei727499@163.com; 5Jilin Jianwei Songkou Biotechnology Co., Ltd., Changchun 510664, China; sengongjianwei@sina.com; 6Benda International Inc., Ottawa, ON K1X 0C1, Canada; chengg_ca@yahoo.com

**Keywords:** taxifolin, LUAD, LUSC, immune escapes, immune checkpoint blockade, overall survival

## Abstract

**Simple Summary:**

Lung cancer (LC) is the leading cause of cancer deaths. While the current immunotherapies are beneficial to patients, the effects are minimal and heterogeneous, which calls for improvements in patient selection and treatment effectiveness. We discovered taxifolin to inhibit LC. The inhibition was associated with alterations of gene expressions. Among those genes affected, a panel of 12 genes, which we named TxflSig (taxifolin signature), and its subpanel of 7 genes (TxflSig1) effectively predicted responses of lung cancer patients to immunotherapy; TxflSig and TxflSig1 are valuable biomarkers for patient selection. Among both TxflSig and TxflSig1 multigene panels are *ITGAL*, *ITGAX*, and *TMEM119* genes. These three genes were robustly associated with immunosuppressive activities, and their expressions were inhibited by taxifolin. Collectively, this research contributes to improvement in the management of lung cancer patients via patient selection and suggests that taxifolin could be a promising addition to immunotherapy in treating LC patients.

**Abstract:**

Using an LL2 cell-based syngeneic mouse LC model, taxifolin suppressed allografts along with the appearance of 578 differentially expressed genes (DEGs). These DEGs were associated with enhancement of processes related to the extracellular matrix and lymphocyte chemotaxis as well as the reduction in pathways relevant to cell proliferation. From these DEGs, we formulated 12-gene (TxflSig) and 7-gene (TxflSig1) panels; both predicted response to ICB (immune checkpoint blockade) therapy more effectively in non-small-cell lung cancer (NSCLC) than numerous well-established ICB biomarkers, including PD-L1. In both panels, the mouse counterparts of *ITGAL*, *ITGAX*, and *TMEM119* genes were downregulated by taxifolin. They were strongly associated with immune suppression in LC, evidenced by their robust correlations with the major immunosuppressive cell types (MDSC, Treg, and macrophage) and multiple immune checkpoints in NSCLC and across multiple human cancer types. ITGAL, ITGAX, and IIT (ITGAL-ITGAX-TMEM119) effectively predicted NSCLC’s response to ICB therapy; IIT stratified the mortality risk of NSCLC. The stromal expressions of ITGAL and ITGAX, together with tumor expression of TMEM119 in NSCLC, were demonstrated. Collectively, we report multiple novel ICB biomarkers—TxflSig, TxflSig1, IIT, ITGAL, and ITGAX—and taxifolin-derived attenuation of immunosuppressive activities in NSCLC, suggesting the inclusion of taxifolin in ICB therapies for NSCLC.

## 1. Introduction

Lung cancer (LC) ranks second in diagnosed cancer incidences in 2020, just behind breast cancer, constituting 11.4% of the 19.3 million new cancer cases. As a leading cause of cancer death, LC contributed 18% of the 10 million cancer fatalities in 2020 [1]. Men have approximately a two-fold higher rate of LC incidence and mortality than women [1]. The disease is composed of two histological types: small-cell lung cancer (SCLC, 15%) and non-small-cell lung cancer (NSCLC, 85%) [2]. The latter contains three subtypes: lung adenocarcinoma (LUAD), lung squamous cell carcinoma (LUSC), and large cell carcinoma, each constituting 50%, 30%, and 10% of NSCLC cases, respectively [3,4,5]. LUAD and LUSC are the main types of NSCLC and the major contributors to LC mortalities. While LUAD and LUSC are driven by activation of the pathways regulating proliferation, cell cycle progression, cell survival, and oxidative stress response, their respective alterations relevant to the pathways are different [6]. Fortunately, in LUAD, mutations occurring in receptor tyrosine kinases (RTK) are targetable, including EGFR (epidermal growth factor receptor), ALK (anaplastic lymphoma receptor tyrosine kinase), and ROS1 (ROS proto-oncogene 1) [6]. The identification of activating mutations in EGFR in 2004 [7,8] transformed the treatment of patients with LUAD from the empirically oriented cytotoxic approach to personalized medicine involving targeted therapies [6,9]. However, as equivalent actionable mutations in RTK rarely occur in LUSC [10], LUSC patients hardly benefit from targeted therapy [11].

The management of NSCLC has shifted owing to the advent of immune checkpoint blockade (ICB) therapy. Since the 2015 FDA approval of nivolumab (anti-PD-1) as a second-line therapy for advanced NSCLCs progressing on previous platinum-based chemotherapy, ICB therapy has gained recognition as a first-line treatment for NSCLCs in advanced and early stages [12]. The latest approval is the use of tremelimumab (anti-CTLA4) in combination with durvaluman (anti-PD-L1) and platinum for patients with metastatic NSCLC, on 10 November 2022 (https://www.fda.gov/drugs/resources-information-approved-drugs/fda-approves-tremelimumab-combination-durvalumab-and-platinum-based-chemotherapy-metastatic-non) (accessed on 12 December 2022). However, only a fraction of NSCLC patients responded to ICB therapy, and many of them experienced disease progression [13,14]. Additionally, ICB treatment can instigate severe immune-related adverse events (irAEs), which may be lethal [12]. PD-L1 expression is the most reliable biomarker currently used in NSCLC, but PD-L1 performed poorly in selecting patients for ICB therapy [12,14]. To improve the management of patients treated with ICB, the development of better biomarkers to stratify patients who will benefit from ICB treatment is a priority.

One of the key pathways altered in NSCLC leads to abnormalities in the oxidative stress response [6]. Taxifolin (dihydroquercetin), a flavonoid with established antioxidant and anti-inflammatory properties [15], might be beneficial in NSCLC therapy. Taxifolin displayed anticancer activities both in vitro and in vivo (xenografts) towards colorectal cancer [16,17,18], liver cancer [19], skin cancer [20], osteosarcoma [21], gastric cancer [22], glioma [23], prostate cancer [24], cervical cancer [25,26], breast cancer [27,28], and lung cancer [29]. The anti-cancer impact of taxifolin was attributable to activation of the Nrf-mediated anti-ROS (reactive oxygen species) cytoprotective pathway, Wnt/β-catenin signaling, and p53 in colon cancer, breast cancer, Ewing’s sarcoma, and cervical cancer [25,30,31]. Given the somatic mutations in TP53 occurring in 90% of LUSC and approximately 50% of LUAD [6], it is appealing to thoroughly investigate taxifolin-derived anti-oncogenic activities towards LUAD and LUSC. This concept is particularly attractive, considering taxifolin’s role as a nutritional supplement and its health-promoting effects [32,33,34].

We report here a systematic investigation of taxifolin-derived anti-NSCLC actions. In our analysis of the mechanisms underlying taxifolin’s inhibition of murine LL2 (Lewis lung carcinoma) cell-produced tumors, which models human NSCLC [35], we detected downregulation of *Itgal*, *Itgax*, and *Tmem119* gene expression among more than 500 differentially expressed genes (DEGs) resulting from taxifolin treatment. In both TCGA LUAD and LUSC cohorts, *ITGAL*, *ITGAX*, and *TMEM119* expressions were robustly correlated with tumor-associated myeloid-derived suppressor cells (MDSCs), regulatory T cells (Tregs), and macrophages, along with the expression of multiple immune checkpoints; these correlations were also detected in a panel of human cancers. ITGAL, ITGAX, the combination of ITGAL-ITGAX-TMEM119 (IIT), and two multigene panels (taxifolin signature/TxflSig and TxflSig1) all effectively predicted the response of NSCLC to anti-PD1-based ICB treatment, which outperformed CD274 (PD-L1)-based prediction. Collectively, this research advances the current understanding of immune escape in NSCLC and may shed light on improving the management of NSCLC patients for ICB therapy.

## 2. Materials and Methods

### 2.1. Data Sources

LUAD and LUSC TCGA datasets organized by cBioPortal [36,37] and the Human Protein Atlas database (https://www.proteinatlas.org/) (accessed on 21 November 2022) were used.

### 2.2. Programs and Websites

UALCAN [38], Metascape [39], and TISIDB [40] (accessed on 2 October 2022) as well as the R packages *dplyr*, *survival*, *Maxstat*, *cutpointr*, and others were utilized in this investigation.

### 2.3. Signature Score Assignment for Individual Tumors

The respective coefficient (coef) for component genes within a multigene panel in predicting the overall survival probability of LC patients was calculated with multivariate Cox regression (the R *Survival* package). Risk scores for every tumor were produced as follows: Sum (coef_1_ x Gene_1exp_ + coef_2_ x Gene_2exp_ + … …+ coef_n_ x Gene_nexp_), where coef_1_ … coef_n_ were the coefs of individual genes and Gene_1exp_ … … Gene_nexp_ were individual gene expressions.

### 2.4. Generation of LL2 Tumors and Taxifolin Treatment

Mouse LL2 LC cells were obtained from ATCC (American Type Culture Collection; ATCC CRL-1642) and cultured in DMEM medium supplemented with fetal bovine serum (FBS, 10%). A mycoplasma PCR detection kit (Abcam, ab289834) was used to monitor mycoplasma contamination. Tumors were generated by subcutaneous (s.c.) implementation of 10^4^ LL2 cells resuspended in 50% Matrigel (Corning 356234) into 6–8-week-old C57BL/6 mice (Jackson 000664). Taxifolin was dissolved in 100 μL PBS mixed with 10% DMSO. The solution was then administered intraperitoneally (ip) into mice on Day 3 after tumor implantation, followed by injections every 3 or 4 days at a dose of 50 mg/kg. Tumor volume was calculated as previously described [41]. Taxifolin was supplied by Jilin Jianwei Songkou Biotechnology Co., Ltd. (Changchun, China) with purity >90% and freshly dissolved in DMSO (dimethyl sulfoxide) at 50 mg/10 μL, followed by dilution in PBS prior to animal application. The proper DMSO + PBS solution was used as the negative control.

### 2.5. RNA Sequencing Analysis

RNA sequencing was carried out as previously published [42] on LL2 tumors treated with either DMSO or taxifolin (*n* = 3 per group). RNA extraction was performed with a miRNeasy Mini Kit (Qiagen, No. 217004) and enriched for poly(A) mRNA using NEBNext^®^ Poly(A) mRNA Magnetic Isolation Modules. Unique dual indexes were used for library preparation, followed by sequencing at the McMaster Genomics Facility using a pair-end, 2 × 50 bp configuration on the Illumina NextSeq 2000 P3 flow cell, with 10 M clusters aimed per sample. Galaxy (https://usegalaxy.org/) (accessed on 21 August 2022) was used to analyze RNA-seq reads, with low-quality reads and adaptor sequences being removed. Alignment to mouse genomic sequence (mm10) was achieved with HISAT2; read counts were executed using the “Featurecounts” function. Differentially expressed genes (DEGs) were produced with DESeq2. KEGG analysis and GSEA (Gene Set Enrichment Analysis) were performed using Galaxy; the FGSEA (fast preranked GSEA) was used for GSEA analysis. Enrichment analyses were carried out using Metascape [39] (https://metascape.org/gp/index.html#/main/step1) (accessed on 1 December 2022) and ClusterProfiler [43].

### 2.6. Immunohistochemistry (IHC)

IHC images to detect ITGAL, ITGAX, and TMEM119 expression in LUAD and LUSC were downloaded from the Human Protein Atlas (https://www.proteinatlas.org/) (accessed on 16 December 2022) using the HPAanalyze R package [44]. The antibodies utilized were CAB025011 for ITGAL, CAB072871 for ITGAX, and HPA051870 for TMEM119.

### 2.7. Statistical Analysis

The R *Survival* package and GraphPad Prism 7 were used to establish Kaplan–Meier survival curves and conduct the logrank test. The R *PRROC* package was utilized to produce ROC (receiver operating characteristic) curves. Mann–Whitney test, Chi-Square test and one-way ANOVA were performed by GraphPad Prism 7 and SPSS 26. Data were presented as mean ± standard deviation (SD). A value of *p* < 0.05 was regarded as statistically significant.

## 3. Results

### 3.1. Taxifolin Inhibits Lung Cancer Growth with a Network Action

Taxifolin’s anti-tumor actions are in accordance with its general health-promoting benefits [32,33,34]. While taxifolin displays a suppressive effect on LC [29], how this tumor inhibitory property functions remains unclear. To fill this knowledge gap, we investigated taxifolin’s impact on LC using likely the only syngeneic mouse NSCLC model available [35], Lewis lung carcinoma, which allows a comprehensive analysis under a fully immunocompetent in vivo environment. We previously optimized experimental conditions on taxifolin-derived inhibition of 4T-1 breast cancer growth in a syngeneic tumor model; intraperitoneal (ip) injection of taxifolin at 50 mg/kg twice per week, which had no impact on weight gain in mice, displayed a clear inhibition of tumor growth [28]. With the optimized conditions, taxifolin significantly delayed LL2 tumor growth in comparison to DMSO treatment (Figure 1A,B). We subsequently profiled gene expressions in tumors treated with taxifolin or DMSO using RNA-seq and observed differences in gene expression caused by taxifolin treatment (Figure 1C).

We then analyzed the pathways or processes enriched in LL2 tumors treated with taxifolin compared to those treated with DMSO. Gene Set Enrichment Analysis (GSEA) using the gene ontology (GO) gene sets revealed that in LL2 tumors, taxifolin treatment resulted in (1) positive enrichment of gene sets regulating extracellular matrix and immunological processes and (2) negative enrichment of pathways regulating cell proliferation, including small G proteins (Rho and Ras), cell proliferation machinery (microtubule nucleation, growth cone, centrosome localization, and spindle pole), and key signaling events (membrane receptor complex, TORC1 signaling, and MAP kinase activity) (Figure 2A,B; Appendix A). Similar enrichments were also obtained using the KEGG gene sets (Appendix A; Appendix A). To analyze the relevance of these enrichments to human LC, we converted the mouse genes to the human homologous genes and performed GSEA with a C8 cell type signature gene set (https://www.gsea-msigdb.org/gsea/msigdb/) (accessed on 10 September 2022); a set of enrichment relevant to lung cells was noted (Appendix A). The above analyses support taxifolin-derived network action in inhibiting LL2 tumor growth, which is likely relevant to human LC pathogenesis.

We further identified taxifolin-induced differentially expressed genes (DEGs) in LL2 tumors (Appendix A), which were defined as *q* < 0.05 and fold change ≥ |1.5| compared to gene expression in DMSO-treated tumors. The positive DEGs (i.e., those DEGs upregulated by taxifolin) were enriched in extracellular matrix– and immune cell–related events (Figure 2C); enrichment by those downregulated DEGs included the processes relevant to G-protein regulation and integrin signaling (Figure 2D), which is highly relevant to the immune escape of LC detected in this study (see the following sections for details). Collectively, the above evidence supports taxifolin-derived suppression of LC pathogenesis through a network, including processes relevant to cell proliferation and immunity.

### 3.2. The Immune Component of Taxifolin-Induced DEGs in Predicting NSCLC Response to ICB Therapy

The PD-1/PD-L1 axis and CTLA4-based ICB therapies constitute the first-line treatment for NSCLC without targeted therapies [11,12]. Given the enrichments in processes relevant to immune regulations in LL2 tumors treated with taxifolin (Figure 2A,C), we analyzed the immune component affected by taxifolin. The murine DEGs (*n* = 578, Appendix A) were converted to the corresponding human genes (*n* = 539, Appendix A) and then screened for their T (cytotoxic T lymphocyte/CTL) cell dysfunction values in five datasets (neuroblastoma, leukemia, breast cancer, endometrial cancer, and melanoma) using the Regulator Prioritization function in the TIDE platform [45,46]. These values reflect individual factors’ involvement in the CTL-mediated killing of tumor cells; for instance, positive and negative T cell dysfunction values indicate the inhibitory and facilitatory impact on CTL’s tumor-cell-killing activities, respectively [45]. We focused on the 12 genes with positive T cell dysfunction values > 0.5 in all five datasets (Figure 3A, top panel); these genes were associated with overall survival (OS) probability in multiple cohorts treated with ICB (Figure 3A, bottom panel). These 12 genes formed a multigene panel TxflSig (taxifolin signature) that predicted response to ICB therapy in multiple cohorts at an AUC (area under the curve) value > 0.5 (Appendix A), with five cohorts (including an NSCLC cohort) showing AUC ≥ 0.7 (Appendix A). The predictions were comparable to some well-established ICB biomarkers (Appendix A). In a NSCLC, melanoma, and glioblastoma cohort, TxflSig exhibited superiority to several effective ICB biomarkers in predicting response to ICB treatments, including TIDE [45], MSI.score (microsatellite instability) [47,48], CD274 (PD-L1) [49], CD8 [50], and INFG (INFγ) [51] (Figure 3B).

The possession of positive T cell dysfunction value for all TxflSig component genes (Figure 3A, top panel) reveals their inhibition of CTL function [45] and potential contributions to immune evasion in cancer. In this regard, the murine counterparts of STRA6, TMEM119, ITGAX, MINK1, NBEAL2, GBP4, and ITGAL genes were downregulated in taxifolin-treated LL2 tumors (Appendix A), which is consistent with their potential roles in inhibiting CTL functions. This suggests a potential biomarker value of these genes in predicting the response of NSCLC to ICB therapy. We named this panel as TxflSig1. In comparison to TxflSig, TxflSig1 displays improved performance as an ICB biomarker (comparing Appendix A to Appendix A). Importantly, TxflSig1 predicted NSCLC response to anti-PD1 therapy at an AUC of 0.8, which was the best among all well-established ICB biomarkers, including CD274, IFNG, and Merck18 (T-cell-inflamed signature) [51] (Figure 3B).

Among the seven component genes of TxflSig1, we recently made several interesting observations of GBP4, including its induction by a gain-of-function PCSK9 mutant in melanoma; its expression being positively associated with T cell dysfunction [52], and its biomarker value in predicting response to ICB therapy (manuscript in preparation). Additionally, we noticed both ITGAL and ITGAX effectively predict NSCLC response to PD1-based ICB treatment as individual genes, with a robustness matching or outperforming other established ICB biomarker-derived predictions (Figure 3C). While TMEM119 displayed modest ICB biomarker potential, the combination of it with ITGAL and ITGAX formed the IIT (ITGAL+ITGAX+TMEM119) panel, with improved ICB biomarker potential in the “Ruppin2021_PD1_NSCLC” cohort organized by TIDE [45] (Figure 3C). Additionally, ITGAL, ITGAX, and IIT possessed effective ICB biomarker potentials towards glioblastoma, melanoma, gastric cancer, and others, with analogous effectiveness compared to well-established ICB biomarkers (Appendix A–F).

To further support the above analyses, TxflSig, TxflSig1, ITGAL, ITGAX, and IIT all predicted OS probabilities in multiple cohorts treated with ICB, including melanoma and clear-cell renal cell carcinoma (ccRCC) (Figure 3D–H and Appendix A). Their positive associations with better prognosis in these patients treated with ICB (Figure 3D–H and Appendix A) suggest their association with CTL dysfunction, as ICB aims to restore CTL function.

In addition to the taxifolin-downregulated genes discussed above, we noticed five genes (Prdx6, Magohb, Nucks1, Dcaf13, and Txn1) being upregulated by taxifolin in LL2 tumors (Appendix A); their human homologous genes, PRDX6, MAGOHB, NUCKS1, DCAF13, and TXN, possessed negative T cell dysfunction values < −1 in all five TIDE datasets (Appendix A), indicating their actions in facilitating CTL function. The magnitude of this facilitation was more prominent than SOX10, a top candidate identified to promote CTL-derived killing of cancer cells [45,53,54]. PRDX6 (peroxiredoxin 6) and TXN (thioredoxin) are known for their antioxidant activities, and TXN enhances NK cell-derived anti-cancer immunity in the tumor microenvironment (TME) [55], suggesting a potential for their inductions to contribute to taxifolin’s antioxidant actions. Although PRDX6, MAGOHB, NUCKS1, DCAF13, and TXN genes as a panel displayed weak prediction of ICB response, the panel robustly stratified overall survival in patients with melanoma treated with anti-PD1 following progression on anti-CTLA4 therapy (Appendix A). Collectively, we provide a comprehensive set of evidence indicating that taxifolin’s suppressive effects on NSCLC are attributable at least in part to the sensitization of LC to immune reactions.

### 3.3. Robust Correlation of ITGAL, ITGAX, and TMEM119 with Immune Checkpoints

Immune checkpoints play critical roles in cancer cells’ evasion of immune surveillance. Of note, using the TCGA Firehose Legacy cohorts within the cBioPortal platform [36,37], we detected high-level correlations of ITGAL with TIGIT, CD96, and BTLA in LUAD (*n* = 517) and with TIGIT, CD96, and PDCD1 in LUSC (*n* = 501) (Figure 4A). High levels of correlation were also observed between ITGAX and CSF1R, HAVCR2, or CTLA4 in both LUAD and LUSC (Figure 4A). We defined high-level correlation at a Spearman correlation (rho) ≥ 0.5. With the setting of ITGAL, ITGAX, or TMEM119 displaying a high-level correlation with immune checkpoints, ITGAL and ITGAX exhibited high levels of correlation with *n* = 18 immune checkpoints in LUAD (Figure 4B); similarly, ITGAL, ITGAX, and TMEM119 correlated with *n* = 19 immune checkpoints in LUSC (Figure 4C). The top correlations of ITGAL with TIGIT and CD96 in both LUAD and LUSC suggest the major immune checkpoints utilized by ITGAL in facilitating immune escape (Figure 4B,C). Similarly, ITGAX may enhance the immunosuppressive actions of HAVCR2 and CSF1R in LUAD and LUSC (Figure 4B,C), while TMEM119 facilitates LUSC immune evasion via CSF1R (Figure 4C). Nonetheless, this does not exclude the network actions of all individual immune checkpoints in ITGAL-, ITGAX-, and TMEM119-derived immune evasion in NSCLC. Furthermore, the high levels of correlation observed above were not limited to NSCLC but across 30 human cancer types, including LUAD and LUSC (Appendix A). Collectively, the above evidence supports the role of taxifolin-downregulated ITGAL, ITGAX, and TMEM119 in enhancing NSCLC immune evasion.

Consistent with this notion, the mouse homologous genes of human PRDX6, MAGOHB, NUCKS1, DCAF13, and TXN were upregulated by taxifolin in LL2 LC tumors (Appendix A) and facilitated CTL-derived toxicity towards cancer cells (Appendix A). In this regard, PRDX6, MAGOHB, NUCKS1, DCAF13, and TXN displayed an overall negative correlation with multiple immune checkpoints in LUAD, LUSC (Appendix A), and other cancer types (Appendix A). These observations further support the impact of taxifolin-affected genes on modulating TME-associated immunity in part via their interactions with immune checkpoints.

### 3.4. A Substantial Correlation of ITGAL, ITGAX, and TMEM119 with LC-Associated MDSC, Treg, and Macrophages

Infiltration of myeloid-derived suppressor cells (MDSCs), regulatory T cells (Treg), and macrophages plays a critical role in shaping immunosuppressive TME [57]. Of note, ITGAL and ITGAX were dramatically correlated with the content of MDSC and Treg in both LUAD and LUSC (Figure 5A). TMEM119 was robustly associated with MDSC and Treg in NSCLC (Figure 4C and Figure 5B). In addition, ITGAL, ITGAX, and TMEM119 were also strongly correlated with macrophages and other immune cell populations in LUAD and LUSC (Figure 4C and Figure 5B) as well as across a panel of human cancer types (Appendix A). However, the strong association with a range of immune cells, including activated CD8 (Act CD8) and NK cells with demonstrated anti-tumor actions, indicates the dynamic nature of TME and, particularly, the continuous actions of the pro- and anti-tumor activities present in TME. In this regard, different B cell populations exist in TME with pro- and anti-tumor activities [58], which is in line with our observations (Figure 5B,C). Alternatively, the infiltrated CD8+ T and NK cells might be dysfunctional or in an anergic state, owing to the high-level correlations of ITGAL, ITGAX, and TMEM119 with immune checkpoints, including TIGIT, CD96, PVRIG, CTLA4, CD274, and others (Figure 4). TIGIT, CD96, and PVRIG are immune checkpoints for both NK and CD8+ T cells [56,59,60,61]. Collectively, the above evidence supports the facilitative roles of ITGAL, ITGAX, and TMEM119 in shaping immunosuppressive TME for LUAD and LUSC.

The above inference is supported by the opposite correlations of PRDX6, MAGOHB, NUCKS1, DCAF13, and TXN genes, the human counterparts of murine genes upregulated in taxifolin-treated LL2 tumors (Appendix A), with NSCLC-associated immune cells. These genes were associated with negative T cell dysfunction value (Appendix A), negatively correlated with the content of MSDC, Treg, and macrophages in both LUAD and LUSC (Figure 4E and Figure 5D), and displayed a general negative association with MSDC, Treg, and macrophages across a set of human cancer types (Appendix A). Taken together, this demonstrates a robust association of ITGAL, ITGAX, and TMEM119 with the major immunosuppressive immune cell types (MDSC, Treg, and macrophages) in LUAD and LUSC, suggesting a contribution of their reductions to taxifolin-derived inhibition of NSCLC pathogenesis.

### 3.5. Expression Status of ITGAL, ITGAX, and TMEM119 in LC

PRDX6, MAGOHB, NUCKS1, DCAF13, and TXN displayed opposite trends on their potential facilitation of CTL function as well as correlations with immune checkpoints and TILs (tumor-infiltrating lymphocytes) compared to ITGAL, ITGAX, and TMEM119 in NSCLC. Nonetheless, their negative correlations were modest compared to the positive correlations displayed by ITGAL, ITGAX, and TMEM119 (comparing Figure 4B,C to Appendix A and comparing Figure 5B,C to Figure 5D,E). Additionally, ITGAL, ITGAX, and TMEM119 effectively predicted NSCLC response to ICB therapy both individually and as a panel (Figure 3C; Appendix A–E), which was not the situation for PRDX6, MAGOHB, NUCKS1, DCAF13, and TXN. We thus focused on a detailed analysis of ITGAL, ITGAX, and TMEM119 expressions in LC.

Our knowledge of the involvement of ITGAL, ITGAX, and TMEM119 in LC is limited. As of 12 August 2023, PubMed listed five articles under the search term of “ITGAL AND Lung cancer”, which studied ITGAL’s biomarker values in NSCLC [62,63,64] and its association with CD4 memory cells in the TME of LUAD [65]. For ITGAX, there were four articles in PubMed documenting its relevance to LUAD, including its expression in dendritic cells (DCs) [66] and its presence in multigene biomarker panels [67,68]. TMEM119 involvement in LC is not clear. Collectively, the expression status of ITGAL, ITGAX, and TMEME119 in NSCLC remains unclear. By using the UALCAN platform (http://ualcan.path.uab.edu/index.html) (accessed on 10 October 2022), we observed a general reduction of ITGAL, ITGAX, and TMEM119 in LUAD at both the mRNA and protein expression levels (Figure 6A–C,H–J) as well as their downregulation in LUSC at the mRNA level (Figure 6K–M). Additionally, further downregulations of ITGAL, ITGAX, and TMEM119 in stages 3 and 4 compared to stage 1 LUAD were observed (Figure 6D,E), and reductions of ITGAX and TMEM119 in stage 4 compared to stage 2 LUAD were also detected (Figure 6E,F). LUAD tumors with lymph node metastasis (N2) displayed lower TMEM119 expression compared LUAD without the metastasis (Figure 6G). For LUSC, further downregulations of ITGAL, ITGAX, and TMEM119 occurred in tumors with TP53 mutations compared to those without TP53 mutations (Figure 6N–P). Collectively, the evidence indicates downregulations of ITGAL, ITGAX, and TMEM119 following NSCLC tumorigenesis and progression.

The above observations, nonetheless, indicate a complex relationship between these factors and NSCLC progression, given their potential contributions to the attenuation of CTL’s actions. To gain further insights into the involvement of ITGAL, ITGAX, and TMEM119 in NSCLC, we analyzed their protein expression. By taking advantage of the protein expression data within the Human Protein Atlas, we observed a general presence of ITGAL and ITGAX in the stromal compartment but not in tumor cells in both LUAD and LUSC (Figure 7A,B, Appendix A, and Appendix A). In some LUSCs, ITGAL expression was present in lymphoid aggregates and tumor necrosis (Figure 7A and Appendix A). Similarly, ITGAX was also detected in immune cells and tumor necrosis in LUAD and LUSC (Figure 7B and Appendix A). In both LUAD and LUSC, the nuclear expression of TMEM119 in tumor cells is apparent (Figure 7C and Appendix A); its presence in stromal and tumor necrosis could also be detected (Figure 7C and Appendix A). In normal lung tissues, TMEM119 is expressed in both the nucleus and cytoplasm (Appendix A). Tumor necrosis promotes metastasis [69] and is associated with poor prognosis [70]; the presence of ITGAL, ITGAX, and TMEM119 in tumor necrosis suggests their roles in facilitating NSCLC progression.

The detection of ITGAL and ITGAX in immune cells was not surprising, given these genes encode for the integrin α L subunit (αL or CD11a) and αX (CD11c) subunit, respectively. Together with the integrin β2 (CD18) subunit, the αLβ2 integrin or lymphocyte function–associated antigen-1 (LFA-1) plays an essential role in T cell migration and activation via interaction with antigen-presenting cells (APCs) [71]; αXβ2 is crucial in myeloid cells’ (neutrophils, monocytes, and dendritic cells) motility and pathogen phagocytosis [72]. TMEM119 is a recently identified marker for microglia, the resident macrophages in the central nervous system (CNS) [73,74]. While limited evidence indicates the contributions of LFA-1, CD11c+ DC, and TMEM11+ microglia to immune tolerance, their involvement in the immune evasion of cancer remains unknown (see Discussion for details).

### 3.6. IIT-Mediated Stratification of NSCLC OS Probabilities

To further define the connection of ITGAL, ITGAX, and TMEM119 to NSCLC, we studied their multigene panel IIT (ITGAL-ITGAX-TMEM119) in predicting the mortality risk of NSCLC. This effort was supported by the formation of the IIT panel in predicting the response of NSCLC to ICB treatment (Figure 3C); the potential of ITGAL, ITGAX, and TMEM119 was further supported by their significant, albeit imperfect, correlations in both LUAD and LUSC (Appendix A). Using the TCGA PanCancer Atlas LUAD dataset, we calculated the IIT risk score for individual tumors using the following formula: ∑(coef_i_ × Gene_iexp_)_n_ (coef_i_: Cox coefficient of gene_i_; Gene_iexp_: expression of Gene_i_; *n* = 3). Coefs were obtained using the multivariate Cox model. Patients in the high-score group were associated with poor mortality (Figure 8A). Similarly, IIT also stratified the mortality risk of patients with LUSC (Appendix A). After adjusting for a set of clinical features including age, sex, M stage, N stage, and T stage, IIT remained a risk factor for fatality in LUAD but not LUSC (Figure 8B; Appendix A). Nonetheless, the inclusion of these clinical features (IIT+Clinicalfeature) significantly improved IIT’s biomarker potential in the stratification of LUAD- and LUSC-associated fatality risk (Figure 8C and Appendix A); the discrimination of the mortality risk by IIT+Clinicalfeature was at ROC-AUC (receiver operating characteristic–area under the curve) values of 0.7 for LUAD (Figure 8D) and 0.61 for LUSC (Appendix A), and PR (precision recall)-AUC values of 0.56 for LUAD (Figure 8E) and 0.53 for LUSC (Appendix A).

We further examined the IIT+Clinicalfeature biomarker potential in LUAD by estimating cutoff points using the empirical, kernel, and normal methods with 1000 bootstraps using the R *cutpointr* package. The median ROC-AUC in both the in-bag and out-of-bag 1000 bootstrapping samples was 0.70 and 0.69, respectively, for the empirical method and 0.7 for the kernel and normal method. With a range of cutoff points estimated by maxstat, empirical, kernel, and normal methods, effective stratification of LUAD mortality was achieved; however, the sensitivity and specificity associated with these cutoff points could be improved (Figure 8F). Nonetheless, the biomarker potential of IIT in the estimation of OS probability of LUAD and LUSC supports the contributions of ITGAL, ITGAX, and TMEM119 to LUAD and LUSC pathogenesis.

## 4. Discussion

Despite the use of targeted therapies and recently established ICB treatments, LC remains the leading cause of cancer deaths. For patients with NSCLCs that do not benefit from targeted treatments (which include almost all LUSCs), ICB treatment is becoming the first-line treatment or standard of care [11,12]. However, only a fraction of patients with NSCLC respond, and a large proportion show disease progression even with treatments [6,11,12]. The situation thus calls for more robust biomarkers in selecting the right patients for ICB therapy and for exploration of more effective ICB options. For both tasks, more novel targets and new ICB therapeutic regimens are needed.

We provide evidence suggesting taxifolin as a potential addition to ICB treatment. Taxifolin significantly delayed the growth of LL2 tumors, a syngeneic model of human NSCLC [35]. One of the network components contributing to taxifolin-derived inhibition of LL2 tumors affected tumor-associated immunity. For instance, the human homologous genes for the murine genes upregulated by taxifolin in LL2 tumors, including *Prdx6*, *Magohb*, *Nucks1*, *Dcaf13*, and *Txn1*, acted to facilitate the CTL-mediated killing of tumor cells, as shown by their negative T cell dysfunction values (Appendix A). On the other hand, *Itgal*, *Itgax*, and *Tmem119* in LL2 tumors were downregulated by taxifolin; their human counterparts (*ITGAL*, *ITGAX*, and *TMEM119*) displayed positive T cell dysfunction values (Figure 3A), suggesting their negative impact on CTL-derived cytotoxicity in tumor cells [45]. These observations are further supported by the negative correlations of *PRDX6*, *MAGOHB*, *NUCKS1*, *DCAF13*, and *TXN* with immune checkpoints (Appendix A) and MDSC, Treg, and macrophages (Figure 5D,E) in comparison to the robust positive correlations of *ITGAL*, *ITGAX*, and *TMEM119* with immune checkpoints (Figure 4) and MDSC, Treg, and macrophages (Figure 5A–C).

The robust and positive correlations of *ITGAL*, *ITGAX*, and *TMEM119* with numerous immune checkpoints, including TIGIT, CD96, PDCD1 (encoding PD-1), CTLA4, LAG3, CD274 (PD-L1), VISTA, ICOS, BTLA, and others, as well as a set of immune cells including MDSC, Treg, and macrophages in NSCLC (Figure 4 and Figure 5B,C) and across human cancer types (Appendix A) strongly suggest the general immunosuppressive actions of *ITGAL*, *ITGAX*, and *TMEM119* in NSCLC and human cancers. This is a novel property for all three genes, although the mechanisms underlying this property remain unknown. LFA-1 (αLβ2)’s adhesion function achieved via binding to ICAM-1 (intercellular adhesion molecule 1) is essential for T cell migration and activation through interaction with APCs [71,75]. In vitro, LFA-1 contributed to the generation of CD4+ T cells with Treg properties [76]. LFA-1 was detected on the MDSC cell surface; evidence also suggests a role of LFA-1 in MDSC generation [77]. In this regard, it can be speculated that in the TME, LFA-1 may be tailored to the generation of Treg and MDSC and thus facilitate immune evasion in NSCLC. This scenario should be further investigated; if confirmed, it would be intriguing to investigate the clinical potential of lifitegrast, an FDA-approved anti-LFA-1 drug that treats ocular inflammation (https://medlineplus.gov/druginfo/meds/a616039.html) (accessed on 15 December 2022) [78], in preventing or hindering NSCLC or other cancers’ immune evasion.

While it remains to be determined whether ITGAL/CD11a associates with immune evasion in NSCLC via LFA-1 or binding to CD18, ITGAL strongly correlates with TIGIT and CD96 at a Spearman rho ≥ 0.8 in both LUAD and LUSC (Figure 4A). Both TIGIT and CD96 inhibit NK and T cell function via binding to CD226 [60], indicating a possibility that ITGAL may suppress NK and CD8+ T cell’s killing activities of tumor cells.

The integrin CD11c/CD18 (αXβ2)-positive DCs are critical for ocular immune privilege (immune tolerance) by promoting Treg development [79]. CD11c+CD11b+DCs enhance T cell tolerance by facilitating Treg generation [80]. However, the functionality of αXβ2 in Treg generation in both studies was unknown. TMEM119 was recently demonstrated to be a specific marker for microglial cells, the resident macrophages in CNS [73,74]. While evidence suggests TMEM119+ microglia contribute to inflammation in the spinal cord [81], their involvement in immune tolerance remains unknown. Collectively, the mechanisms underlying ITGAX- and TMEM119-derived facilitation of the immunosuppressive TME require further investigations.

Regardless of what might be the mechanisms governing ITGAL, ITGAX, and TMEM119’s contribution to immune evasion in NSCLC and other cancers, their strong associations with immunosuppressive activities (immune checkpoints, MDSC, and Treg) underlie their ICB biomarker values and thus patient selection for ICB therapy, i.e., tumors expressing high levels of these proteins are likely to respond to ICB. Indeed, ITGAL, ITGAX, and the panel of these three genes (IIT) estimate the response of NSCLC to anti-PD1 therapy more efficiently than the panel of well-established ICB biomarkers, including PD-L1 (CD274), INFγ, CD8, and Merck18 (Figure 3C). This level of robustness is also reflected in the performance of TxflSig and TxflSig1 (Figure 3B). Considering the unmet need for better ICB biomarkers in NSCLC, the novel ICB biomarkers identified in this study may improve the stratification of NSCLC patients that can benefit from ICB treatments. However, given the retrospective nature and small cohort sizes of the datasets available on the TIDE platform [45], the ICB biomarkers reported in this investigation will require further examination.

Our research suggests a potential role for taxifolin in improving ICB therapy for NSCLC. Nonetheless, taxifolin (dihydroquercetin/DHQ) possesses poor water solubility and thus limited bioavailability, which may hinder its clinical potential. To address this challenge and enhance taxifolin’s clinical application, improvement of its water solubility is warranted. Towards this end, chemical modification of taxifolin to yield aminomethylated DHQ (DHQA) [82] and complexing taxifolin (TAX) into nanoparticles to produce water-soluble or aqua taxifolin (aqTAX) [83,84] can substantially alleviate the water solubility challenge. Both DHQA and aqTAX either maintain or enhance taxifolin’s pharmacological properties, including its antioxidant potential. Of importance, aqTAX displays enhanced in vivo protection of cortical ischemia in mice compared to taxifolin, even at lower doses [84,85]. It is thus an appealing possibility that DHQA and aqTAX may deliver a strong attenuation of the immunosuppressive feature of NSCLC, which may allow DHQA and aqTAX to facilitate ICB therapy more effectively than taxifolin in NSCLC. This scenario should be investigated in future.

## 5. Conclusions

We report here multiple discoveries with potential impacts on NSCLC therapy and management. (1) Taxifolin inhibits NSCLC via network action, including downregulation of TME-associated immune evasion capacity. This opens the possibility of including taxifolin in the current regimen of ICB therapy in NSCLC. This possibility can be readily tested in clinical trials, owing to taxifolin’s status as a nutritional supplement. (2) Several novel biomarkers in predicting NSCLC response to ICB therapy were identified, including ITGAL, ITGAX, IIT, TxflSig, and TxflSig1. Importantly, they demonstrated higher robustness compared to not only the current ICB biomarker (PD-L1) used in NSCLC patients, but also those of well-studied pan-cancer ICB biomarkers. Thus, these panels’ ICB biomarker potential, particularly in LC, is worthy of further examination. (3) NSCLC and other human cancers may utilize ITGAL, ITGAX, and TMEM119 to shape the TME into an immunosuppressive TME. This observation is novel; the underlying mechanisms should be further investigated to deepen the current understanding of NSCLC- and cancer-associated immune evasion. (4) We provide the first evidence for the role of TMEM119 in NSCLC. Collectively, this research possesses clinical and basic research potential in improving the management of patients with NSCLC.

## Figures and Tables

**Figure 1 cancers-15-04818-f001:**
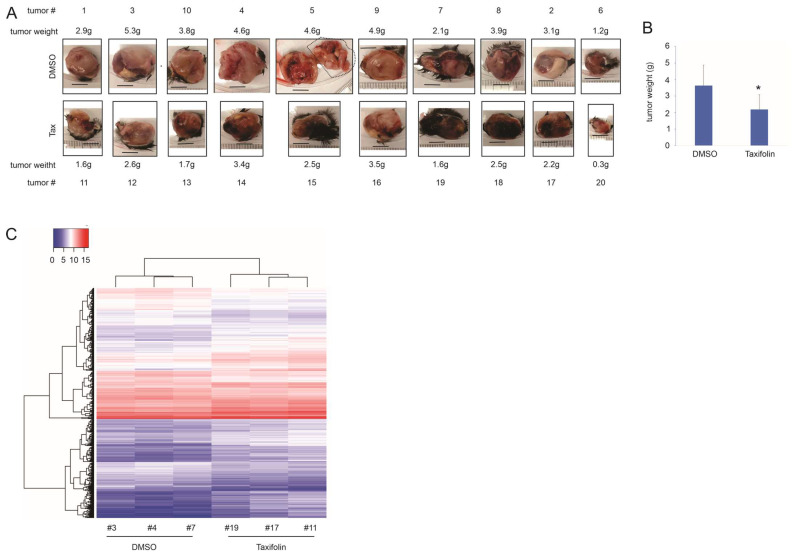
Taxifolin-derived inhibition of LL2 lung tumor growth. (**A**,**B**) LL2 tumors were produced in C57BL/6 mice, treated with either DMSO or taxifolin, and harvested on day 28 post-tumor implantation. Images and tumor weights of individual tumors are shown. The marked tumor portion of DMSO tumor #5 was from a peritoneal metastasis mass (**A**). Quantification was provided with statistical analysis performed using 2-tailed Student’s *t*-test, * *p* < 0.05 (**B**). (**C**) RNA-seq analysis of LL2 tumors treated with DMSO (*n* = 3) or taxifolin (*n* = 3). Heatmap is generated for DEGs derived from the indicated treatments.

**Figure 2 cancers-15-04818-f002:**
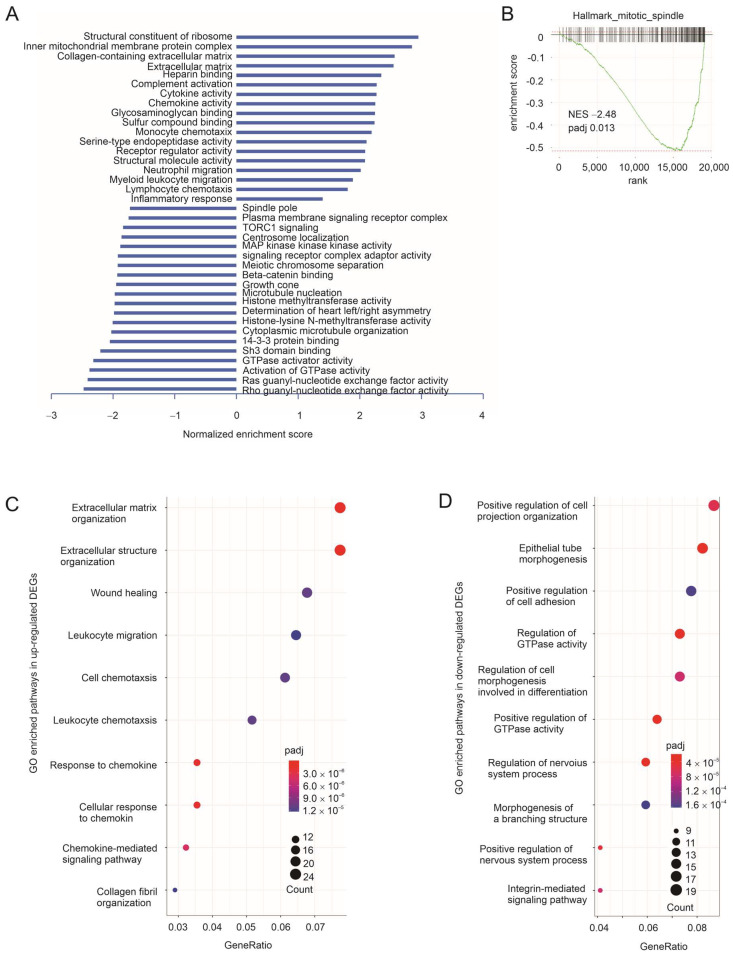
Pathways enriched in LL2 tumors treated by taxifolin. (**A**) Summary of GO GSEA analysis using the gene expression profile of LL2 tumors treated with DMSO or taxifolin. Enrichment analysis was performed using the R package of ClusterProfiler [43]. (**B**) A typical GSEA analysis using the Hallmark gene set within the MSigDB gene set collection; the analysis was performed using Galaxy. NES: normalized enrichment score. (**C**,**D**) GO enrichment analyses were performed using upregulated and downregulated DEGs in LL2 tumors treated with taxifolin.

**Figure 3 cancers-15-04818-f003:**
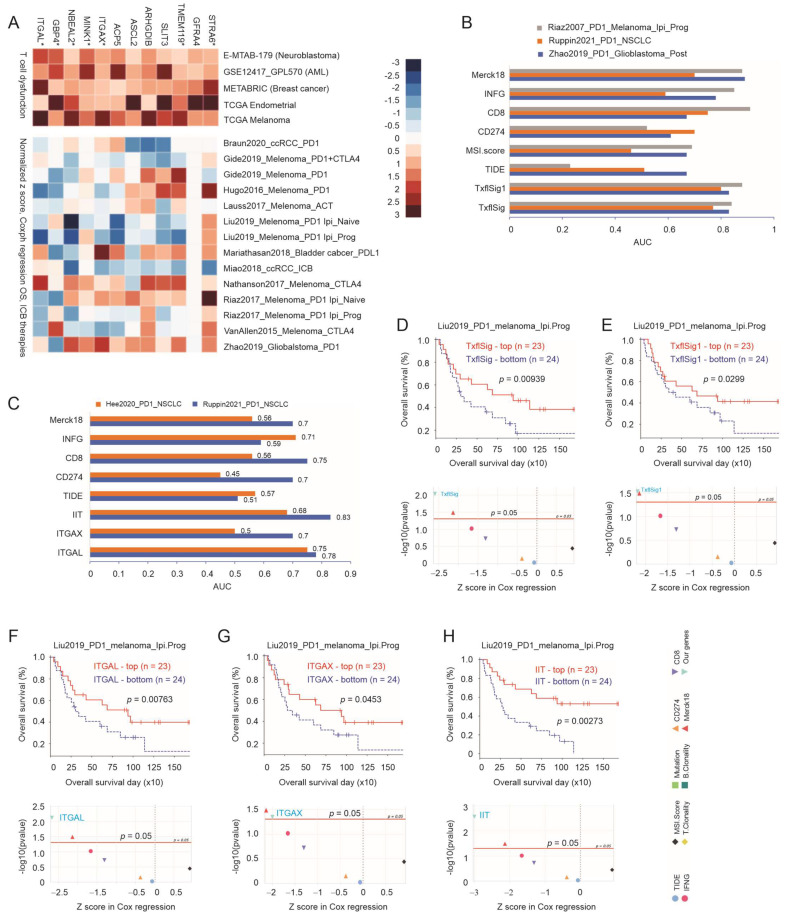
Taxifolin DEGs predict NSCLC response to ICB therapy. (**A**) Association of the indicated DEGs with T cell dysfunction value (top panel) and overall survival in the indicated cancer populations treated with the indicated ICB (bottom panel). Ipi: Ipilumuman anti-CTLA4; Prog: progressed; PD1.Ipi_Prog and PD1.Ipi_Naive: treatment of melanoma by anti-PD1 on tumor progressed or naïve on Ipi. *: the mouse homologous genes were downregulated in LL2 tumors treated with taxifolin; these genes formed a multigene panel TxflSig1, while all 12 genes were formulated to TxflSig. (**B**) Prediction of response to ICB therapies in NSCLC and other indicated cancer cohorts by TxflSig and TxflSig1 along with a set of published ICB biomarkers. The analysis was performed using the TIDE platform with cohort details listed in [45]. (**C**) Prediction of response to ICB therapies in two NSCLC datasets using the TIDE platform with ITGAL, ITGAX, IIT (ITGAL+ITGAX+TMEM119), and other well-studied ICB biomarkers. (**D**–**H**) Stratification of OS in the indicated cancer cohorts treated with the indicated ICB therapies by TxflSig, TxflSig1, ITGAL, ITGAX, and IIT. Top panel: Kaplan–Meier survival curves for the indicated biomarkers; bottom panel: summary of mortality risk stratified by our biomarkers along with a set of well-studied ICB biomarkers. Please note the superiority of our factors in the risk stratification compared to those of published ICB biomarkers.

**Figure 4 cancers-15-04818-f004:**
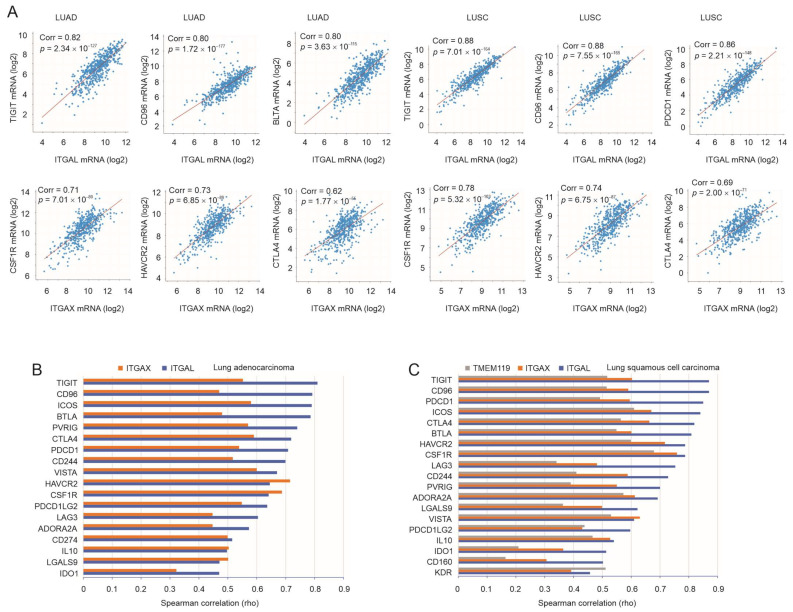
Robust correlation of ITGAL, ITGAX, and TMEM119 with multiple immune checkpoints. (**A**) Spearman correlations (Corr) of ITGAL and ITGAX with the indicated immune checkpoints in LUAD (*n* = 510) and LUSC (*n* = 501) were determined using cBioPortal with the tools provided. (**B**,**C**) Systemic analyses of Spearman correlation between ITGAL, ITGAX, or TMEM119 and the indicated immune checkpoints in LUAD and LUSC were performed with the TISIDB platform [40], except PVRIG (poliovirus-related immunoglobulin domain-containing), an immune checkpoint [56], which was determined using cBioPortal. These immune checkpoints are presented if Spearman rho is ≥0.5 for one of ITGAL, ITGAX, and TMEM119.

**Figure 5 cancers-15-04818-f005:**
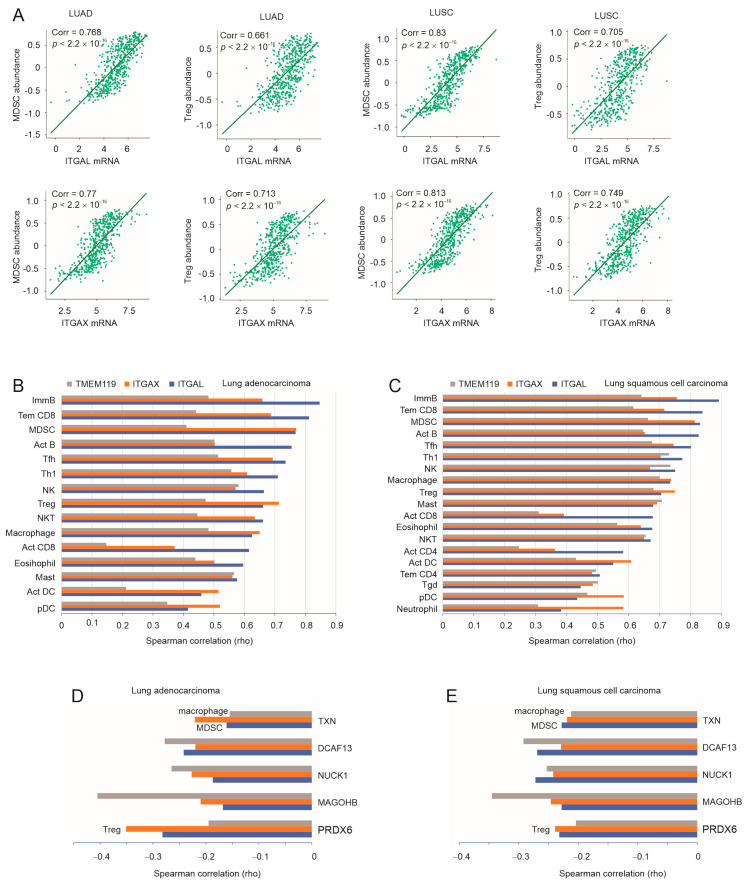
Association of taxifolin-affected DEGs with immune cells infiltrated in LUAD and LUSC. (**A**) Spearman correlations of ITGAL and ITGAX with the content of MDSC and Treg in LUAD and LUSC were obtained using TISIDB. (**B**,**C**) Systemic analyses of Spearman correlations of ITGAL, ITGAX, and TMEM119 with a set of immune cells within the TISIDB platform. Individual immune cell types are included if the correlation for one of the three factors is ≥0.5. (**D**,**E**) Spearman correlations of the indicated genes with Treg, MDSC, and macrophages in LUAD and LUSC.

**Figure 6 cancers-15-04818-f006:**
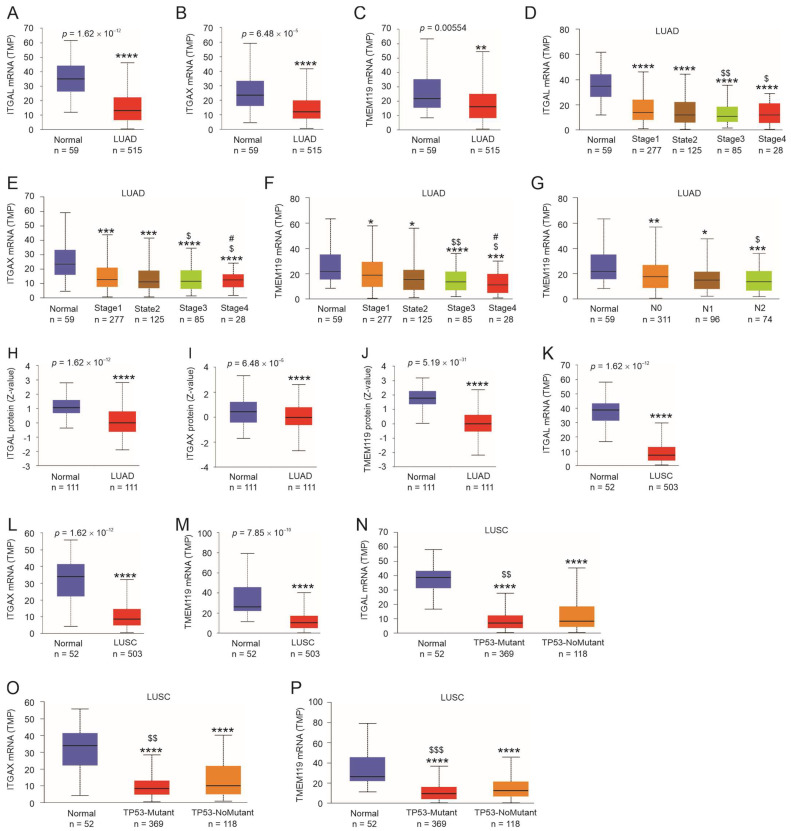
Expression of ITGAL, ITGAX, and TMEM119 in LUAD and LUSC. Analyses were performed using the TCGA dataset organized by UALCAN [38]. *, **, ***, and ****: *p* < 0.05, 0.01, 0.001, and 0.0001, respectively, in comparison to normal tissues; $, $$, and $$$: *p* < 0.05, 0.01, and 0.001, respectively, in comparison to stage 1 tumors (**A**–**F**,**H**–**M**), N0 tumors (**G**), or tumors without TP53 mutation (**N**–**P**); #: *p* < 0.05 compared to stage 2 tumors (**E**,**F**).

**Figure 7 cancers-15-04818-f007:**
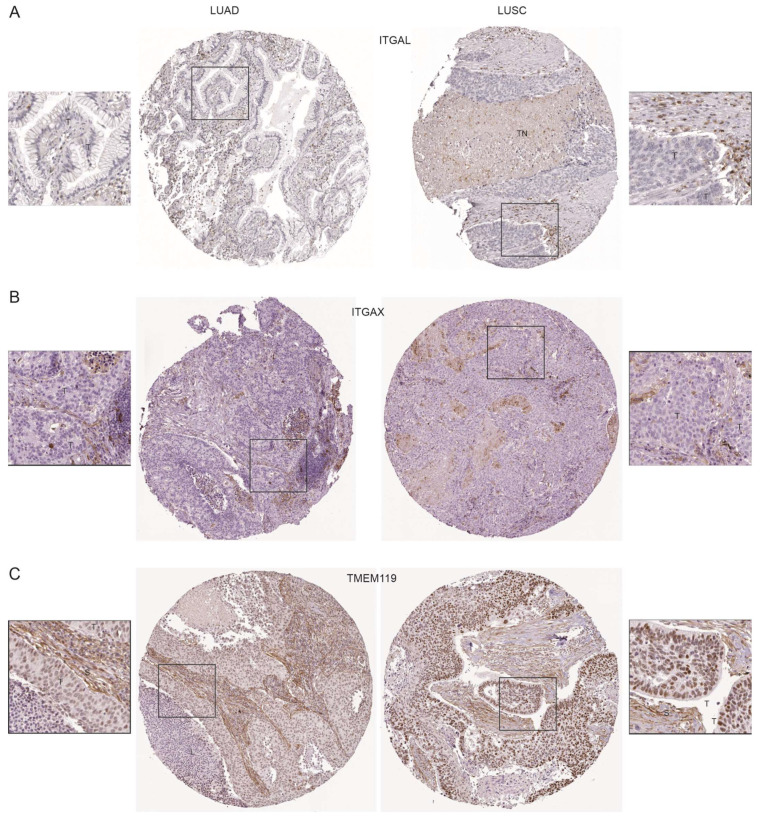
ITGAL, ITGAX, and TMEM119 protein expression in LUAD and LUSC. Immunohistochemistry (IHC) staining of LUAD and LUSC tissue microarray for ITGAL (**A**), ITGAX (**B**), and TEME119 (**C**) protein was downloaded from the Human Protein Atlas (https://www.proteinatlas.org/) (accessed on 12 December 2022). The marked regions were enlarged 2.2-fold. T: tumor cells; TN: tumor necrosis; L: immune cells; S: stroma.

**Figure 8 cancers-15-04818-f008:**
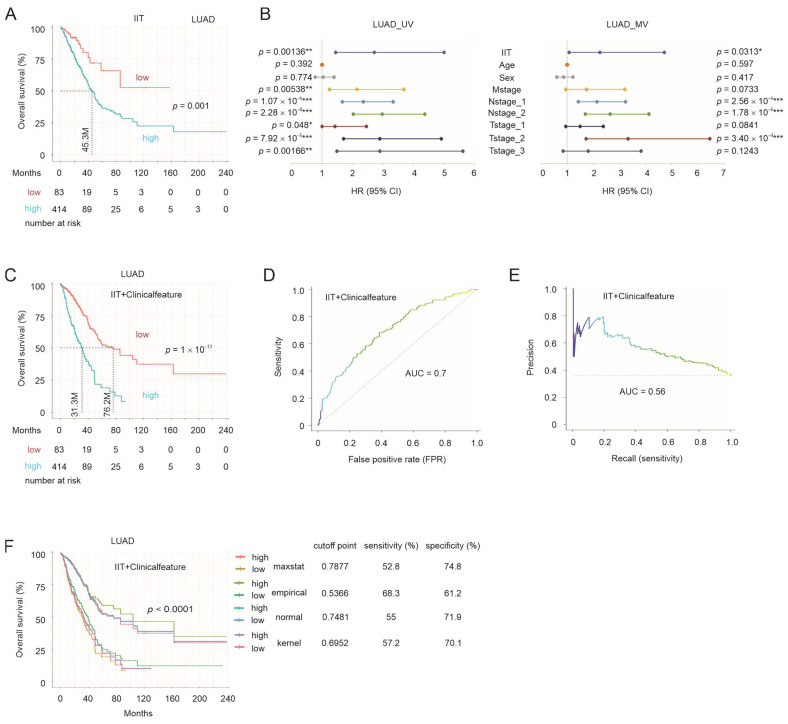
Prediction of OS by IIT. (**A**) The risk score of IIT (ITGAL+ITGAX+TMEM119) was calculated for individual tumors within the TCGA PanCancer Atlas LUAD dataset, followed by estimation of the cutoff point by Maximally Selected Rank Statistics with the R Maxstat package, construction of Kaplan–Meier curves, and log-rank test analysis. (**B**) Univariate (UV) and multivariate (MV) Cox analysis of LUAD mortality risk using IIT and the indicated clinical features. Sex: men compared to women; Mstage: M1 versus M0; Nstage: 1 (N1) and 2 (N ≥ 2) compared to N0; Tstage: T1, T2, and T3 (T3 + T4) compared to T0. *, **, and ***: *p* < 0.05, 0.01, and 0.001 respectively. (**C**) Stratification of mortality risk by IIT together with the above clinical features (IIT+Clinicalfeature). (**D**,**E**) Discrimination of OS by IIT+Clinicalfeature with ROC-AUC (receiver operating characteristic–area under the curve) and precision recall (PR)-AUC curves for LUAD. (**F**) Cutoff points of IIT+Clinicalfeature risk score in stratification of LUAD were estimated by the maxstat, empirical, normal, and kernel methods, followed by the construction of individual survival curves; the individual cutoff points and their associated specificity and sensitivity are presented.

## Data Availability

All materials are available upon request.

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
