# Peer review of "Taxifolin Inhibits the Growth of Non-Small-Cell Lung Cancer via Downregulating Genes Displaying Novel and Robust Associations with Immune Evasion Factors"

_cancers, 2023, doi:10.3390/cancers15194818_

Round 1
Reviewer 1 Report
This is an interesting study showing original findings of biomarkers for NSCLC. Even though there are some limits, acknowledged by the authors, it is an interesting contribution overall.
There are some typographical errors within the figures and also in the manuscript. Therefore I urge the authors to do careful and extensive checking.
I have no other comments and recommend to accept the manuscript.
Overall the language is fine, however, I recommend minor editing.
Author Response
We appreciate the reviewer’s positive comments. Here are our detailed revisions.
“There are some typographical errors within the figures and also in the manuscript. Therefore I urge the authors to do careful and extensive checking.”
“Overall the language is fine, however, I recommend minor editing.”
Authors' response – We appreciate the reviewer’s comment and have diligently reviewed and corrected all topographical errors in both the manuscript and figures to enhance the overall quality of our work.
Reviewer 2 Report
1) NSCLC – the title of the article should be written in full. This abbreviation must be deciphered the first time it is mentioned. 2) Figure 1 is difficult to read. These figures should be divided into 2 figures 3) The methods do not contain a description of Immunohistochemistry (IHC) staining 4) The authors should discuss possible forms of taxifolin, for example as a water-soluble complex or nanoparticles. A Comparative Analysis of Neuroprotective Properties of Taxifolin and Its Water-Soluble Form in Ischemia of Cerebral Cortical Cells of the Mouse - PubMed (nih.gov) Cytoprotective Properties of a New Nanocomplex of Selenium with Taxifolin in the Cells of the Cerebral Cortex Exposed to Ischemia/Reoxygenation - PubMed (nih.gov)
The quality of English is acceptable. It is necessary to additionally check the text for typos
Author Response
We greatly appreciate the reviewer’s remarks. Here are our detailed revisions.
1) “NSCLC – the title of the article should be written in full. This abbreviation must be deciphered the first time it is mentioned.”
Authors' response – We have spelled out the full term of Non-Small Cell Lung Cancer in the title.
2) “Figure 1 is difficult to read. These figures should be divided into 2 figures”
Authors' response – We agree! In response to this suggestion, Figure 1 has been divided into two separate figures based on content. Figure 1 now highlights the impact of taxifolin on LL2 tumors with respect to tumor growth and gene expression, while Figure 2 focuses on pathway or process alterations resulting from taxifolin’s effects on gene expression. The number for the rest of figures has been accordingly revised. This revision improved the data presentation, to which we thank the reviewer for his/her comments.
3) “The methods do not contain a description of Immunohistochemistry (IHC) staining”
Authors' response – We appreciate the reviewer for highlighting this omission. The IHC images related to ITGAL, ITGAX and TMEM119 expressions in lung adenocarcinoma (LUAD) and lung squamous cell carcinoma (LUSC) were obtained from the Human Protein Atlas. Detailed information regarding the program used for image downloading, and the antibodies used in IHC staining have now been organized and presented in “Materials and Methods” (see section 2.6, lines 153-157, page 4, and marked with red).
4) “The authors should discuss possible forms of taxifolin, for example as a water-soluble complex or nanoparticles. A Comparative Analysis of Neuroprotective Properties of Taxifolin and Its Water-Soluble Form in Ischemia of Cerebral Cortical Cells of the Mouse - PubMed (nih.gov) Cytoprotective Properties of a New Nanocomplex of Selenium with Taxifolin in the Cells of the Cerebral Cortex Exposed to Ischemia/Reoxygenation - PubMed (nih.gov)”
Authors' response – We greatly appreciate these insightful remarks! Indeed, poor water solubility and the resulting low level of bioavailability for taxifolin were major concerns in its in vivo applications. The chemical modification introduced into taxifolin and complexing taxifolin into nanoparticles have produced aminomethylated DHQ (taxofilin) or DHQA and water-soluble or aqua taxifolin (aqTAX), respectively. DHQA and aqTAX either maintain or enhance taxifolin’s pharmacological properties, including its antioxidant potential. Of importance, aqTAX displays enhanced in vivo protection of cortical ischemia in mice compared to taxifolin even at lower doses. This knowledge is the core content of the literature suggested by Reviewer #2. The readily water soluble DHQA and aqTAX suggest that they may enhance taxifolin’s property in reducing NSCLC-associated immune evasion and thus better facilitate ICB therapy in NSCLC. We have fully discussed this potential and cited all relevant references, as suggested by the review, in this revision (see the new paragraph added, lines 559-572, page 20). We acknowledge that these discussions strengthen the theme of this study, to which we thank Reviewer #2 for the comments.
Other comments
“The quality of English is acceptable. It is necessary to additionally check the text for typos”
Authors' response – We have thoroughly edited the manuscript and trust this improved the manuscript.